

# Effect of genetic background on the evolution of Vancomycin-Intermediate *Staphylococcus aureus* (VISA)

Michelle Su[1], Michelle H. Davis[1], Jessica Peterson[1], Claudia Solis-Lemus[2], Sarah W. Satola[1] and Timothy D. Read[1,3]

[1] Division of Infectious Diseases, Department of Medicine, School of Medicine, Emory University, Atlanta, Georgia, USA
[2] Wisconsin Institute for Discovery and Department of Plant Pathology, University of Wisconsin-Madison, Madison, Wisconsin, USA
[3] Department of Dermatology, School of Medicine, Emory University, Atlanta, Georgia, USA

## ABSTRACT

Vancomycin-intermediate *Staphylococcus aureus* (VISA) typically arises through accumulation of chromosomal mutations that alter cell-wall thickness and global regulatory pathways. Genome-based prediction of VISA requires understanding whether strain background influences patterns of mutation that lead to resistance. We used an iterative method to experimentally evolve three important methicillin-resistant *S. aureus* (MRSA) strain backgrounds—(CC1, CC5 and CC8 (USA300)) to generate a library of 120 laboratory selected VISA isolates. At the endpoint, isolates had vancomycin MICs ranging from 4 to 10 µg/mL. We detected mutations in more than 150 genes, but only six genes (already known to be associated with VISA from prior studies) were mutated in all three background strains (*walK, prs, rpoB, rpoC, vraS, yvqF*). We found evidence of interactions between loci (e.g., *vraS* and *yvqF* mutants were significantly negatively correlated) and *rpoB, rpoC, vraS* and *yvqF* were more frequently mutated in one of the backgrounds. Increasing vancomycin resistance was correlated with lower maximal growth rates (a proxy for fitness) regardless of background. However, CC5 VISA isolates had higher MICs with fewer rounds of selection and had lower fitness costs than the CC8 VISA isolates. Using multivariable regression, we found that genes differed in their contribution to overall MIC depending on the background. Overall, these results demonstrated that VISA evolved through mutations in a similar set of loci in all backgrounds, but the effect of mutation in common genes differed with regard to fitness and contribution to resistance in different strains.

## INTRODUCTION

Vancomycin has been a relatively safe and economical drug against MRSA (methicillin-resistant *Staphylococcus aureus*) for the past three decades but there have been numerous reports of strains with reduced susceptibility (which we will call "vancomycin-resistant", using the common terminology) (*Gardete & Tomasz, 2014*; *Zhang et al., 2015*; *McGuinness, Malachowa & DeLeo, 2017*). High-level vancomycin-resistant *S. aureus*

Corresponding author
Timothy D. Read, tread@emory.edu

(VRSA; minimum inhibitory concentrations (MIC) ≥ 16 µg/mL) strains have acquired the *vanA* gene from horizontal transfer from *Enterococcus* spp (*Kobayashi, Musser & DeLeo, 2012*). The incidence of VRSA has remained rare, likely due to poor regulation of expression of the *vanA* gene in *S. aureus* imposing a fitness burden on the bacterium (*Foucault, Courvalin & Grillot-Courvalin, 2009*). Vancomycin-intermediate (VISA; MIC 4–8 µg/mL) strains, first reported in 1997 (*Hiramatsu et al., 1997*), are more commonly encountered in the clinic than VRSA. VISA evolves from vancomycin-susceptible *S. aureus* (VSSA; MIC ≤ 2 µg/mL) through intrinsic mutagenesis. VISA strains susceptible in vitro to vancomycin (MIC ≤ 2 µg/mL) but containing subpopulations that can grow in the presence of ≥ 4 µg/mL of vancomycin and thus are capable of spontaneous transitions to VISA are termed hVISA (for heterogeneous VISA) (*Liu & Chambers, 2003*; *Sakoulas & Moellering, 2008*; *Deresinski, 2009*; *El-Halfawy & Valvano, 2015*).

Most intermediate resistance is acquired by within-patient evolution (*Klevens et al., 2007*; *Gardete & Tomasz, 2014*). The classic genomics study by *Mwangi et al. (2007)* showed VISA evolving within a patient on long-term vancomycin therapy through a series of adaptive mutations. VISA has emerged independently in strains from each of the major MRSA lineages (*Hiramatsu et al., 1997*; *Howden et al., 2006*, *2008*; *Klevens et al., 2007*; *Mwangi et al., 2007*; *Alam et al., 2014*; *Gardete & Tomasz, 2014*). Some patients with MRSA infections failing therapy showed enhanced vancomycin MIC even when the drug was not used (*Horne et al., 2009*; *Lalueza et al., 2010*; *Holland & Fowler, 2011*), suggesting that VISA may overlap with 'persister' phenotypes (*Johnson & Levin, 2013*) associated with long-term invasive infection.

Standard microbiology laboratory-based phenotypic tests for VISA/hVISA are labor intensive, and the incidence may be higher than currently reported (*Marlowe et al., 2001*; *Charles et al., 2004*; *Prakash, Lewis & Jorgensen, 2008*; *Satola et al., 2009*; *Swenson et al., 2009*; *Vaudaux et al., 2010*). hVISA strains are particularly challenging to detect using current standard clinical microbiology methods (automated broth microdilution assays) as MICs can overlap with VSSA (*Swenson et al., 2009*). Development of a nucleic acid test using standard PCR based approaches has been confounded to date by the complexity of the genetics of VISA. The genetic basis of VISA has been investigated for several years, primarily through identification of mutations by comparative sequencing of small numbers of isogenic clinical samples (*Ohta et al., 2004*; *Mwangi et al., 2007*; *Howden et al., 2008*) and molecular genetic characterization. VISA cells typically show cell-wall thickening and a reduction of fitness in growth compared to isogenic parents. Mutations in a variety of conserved core *S. aureus* genes have been reported to be associated with VISA (*Howden et al., 2010*; *Gardete & Tomasz, 2014*; *Wang et al., 2016*), most commonly those involved in regulation of cell wall architecture (e.g., *graRS* (*Neoh et al., 2008*; *Howden et al., 2008*), *vraRS* (*Kato et al., 2010*; *Baek et al., 2017*; *Asadpour & Ghazanfari, 2019*), *yvqF* (*Kato et al., 2010*; *Yoo et al., 2013*), *walK* (*Shoji et al., 2011*; *Hafer et al., 2012*; *McEvoy et al., 2013*; *Hu et al., 2015*), *walR* (*Howden et al., 2011*)) or certain global transcriptional regulators (e.g., *agr* (*Sakoulas et al., 2003*), *stp1* (*Cameron et al., 2012*; *Passalacqua et al., 2012*), *rpoB* (*Cui et al., 2010*; *Watanabe et al., 2011*; *Hafer et al., 2012*; *Katayama et al., 2017*) and *rpoC* (*Matsuo et al., 2013*)). *walK* controls cell wall autolysis while *vraS* and *yvqF*

(also known as *vraT*) regulate cell wall synthesis, thus mutations in these genes contribute to the characteristic thick cell wall of VISA strains.

It is vital for any future gene-based test for VISA to know if there are epistatic interactions between *S. aureus* strain background and drug resistance mutations. Recent publications have reported important epistatic interactions that drive patterns of antibiotic-resistance (*Schubert et al., 2018*; *Ma et al., 2020*). There are also examples of strain-specific effects in *S. aureus*: clinical CC30 *S. aureus* strains were found to have elevated (average 100 fold higher) persister formation compared to CC5, CC8, CC30, and CC45 strains (*Liu et al., 2020*), and under experimental evolution, CC398 strains were found to frequently evolve ciprofloxacin resistance due to a lineage-specific IS element that allowed amplification of the *norA* efflux pump (*Papkou et al., 2020*). In this study, we aimed to evaluate the relationship between genetic determinants of intermediate vancomycin resistance and strain background to determine if epistasis can direct the evolutionary trajectory of this phenotype. We evolved VISA strains from three common genetic backgrounds encountered clinically and analyzed the differences in mutation, fitness, and resistance levels.

## MATERIALS & METHODS

### Strains
Strains used for the evolution experiments: NRS70 (N315), NRS123 (MW2), and NRS384 (USA300-0114), were obtained from BEI resources (https://www.beiresources.org/).

### Experimental evolution of vancomycin resistance
The strategy for the evolution experiment is outlined in Fig. 1. Parent strains were streaked on Brain Heart Infusion (BHI) plates from frozen stocks and single colonies used to establish shaking overnight cultures of BHI broth. We then propagated independent lines in BHI broth shaking at 250 rpm at 37 °C until cultures were turbid (OD600 > 1.5) before diluting cultures at a 1:500 dilution into fresh media. Strains were grown initially on BHI broth containing 1 μg/mL vancomycin then transferred to BHI broth with 2 μg/mL antibiotic. Once turbid, the cultures were similarly diluted to 4 μg/mL, then grown for 3 days before transfer to 8 μg/mL, and grown for a final 3 days. After growth in broth at the final antibiotic concentration, cultures were plated on BHI plates containing 2, 4, and 8 μg/mL of vancomycin. A single colony was picked from the highest concentration at 48 h growth. The colony was passaged twice on selective media and then examined by Gram stain to confirm species.

### Genomic analysis
DNA extraction and library prep were performed as manufacturer's instructions (Wizard Genomic DNA Purification Kit; Promega, Madison, WI, USA; Nextera XT DNA Library Prep Kit; Illumina, San Diego, CA, USA). Genome sequencing was performed on Illumina HiSeq and MiSeq with paired-end reads. Raw read data were deposited in the NCBI Short Read Archive under project accession number PRJNA525705.

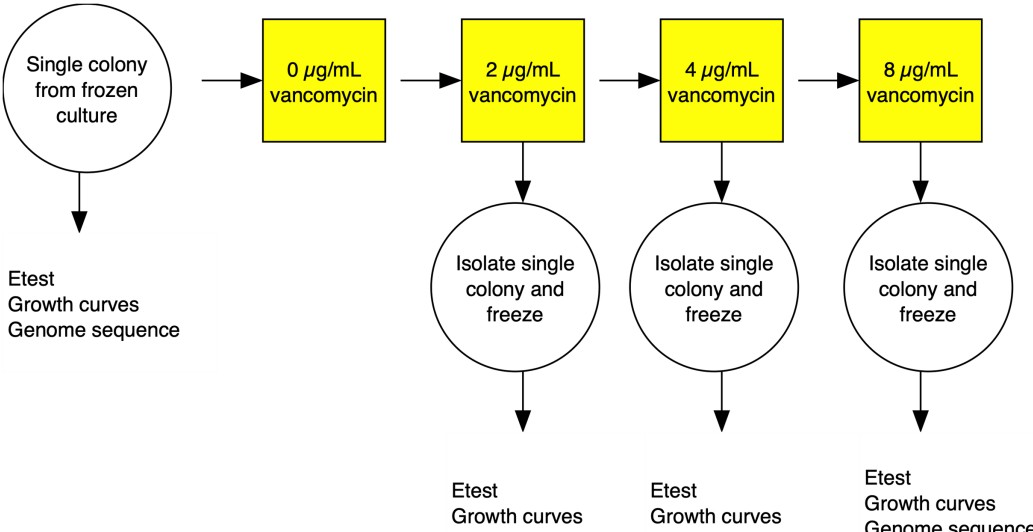

**Figure 1 Experimental evolution of a lineage VISA.** For each of the three *S. aureus* genetic backgrounds, up to 40 independent lineages were evolved from single colonies isolated from frozen culture streaks. The lineages were cultured in BHI broth containing increasing concentrations of vancomycin. At each stage, single colonies were isolated and frozen and tested for antibiotic MIC using Etest strips (vancomycin and daptomycin) and fitness using a growth curve based assays. Full details in the materials and methods section.

Raw read data was processed by BBduk (v37.66) (*Bushnell, 2016*) to filter out adapters associated with Illumina sequencing and trim reads based on quality. Mutations were called based on comparison to reference sequences using *breseq* (*Deatherage & Barrick, 2014*) CONSENSUS mode. Reference genomes used for NRS70, NRS123, and NRS384 VISA strains were NC_002745.2 (BioProject: PRJNA224116, BioSample: SAMD00061099, Assembly: GCF_000009645.1), NC_003923.1 (BioProject: PRJNA224116, BioSample: SAMD00061104, Assembly: GCF_000011265.1), and NZ_CP027476.1 (BioProject: PRJNA224116, BioSample: SAMN07411405, Assembly: GCF_002993865.1) respectively. To compare mutations across the three strains, the protein sequences of NRS123 and NRS384 were aligned using BLAST against that of NRS70 (*Camacho et al., 2009*). Genes were considered orthologous to NRS70 if the amino acid sequences had at least 95% identity, 95% coverage, or an evalue = 0. To determine if large scale deletions were due to phage loss, PhageWeb was used to identify prophages within the genomes (*Sousa et al., 2018*).

## Measuring fitness based on growth

Strains were grown in duplicate in a 96-well plate beginning at an OD < 0.1 and grown for 24 h at 37 °C with constant shaking in a Biotek Eon Microplate Spectrophotometer, with OD measurements every 10 min. To assess growth curves, OD readings were imported into R (*R Core Team, 2016*) and maximal growth rate (*r*) calculated using the *growthcurver* package (*Sprouffske & Wagner, 2016*) using the average of two biological replicates. Fitness was calculated as the ratio between the average *r* of each evolved strain to the average *r* of the parent strain. Fitness distributions between NRS70 evolved VISA strains and

NRS384 evolved VISA strains were compared using a two-sided two-sample Kolmogorov-Smirnov test.

## Statistical analysis

To determine if the difference in the prevalence of mutations across different VISA genes (*walK, prs, rpoB, rpoC, vraS, yvqF*) was significant, a Pearson's Chi-squared test was performed, with subsequent post hoc pairwise comparisons using Fisher's exact test and a Bonferroni multiple test correction. To analyze if the prevalence of mutations was also affected by background, binomial generalized linear models (GLMs) were fitted for each gene, and pairwise Wald tests were performed to test whether the difference in the coefficients for each background was non-zero. The statistical significance of gene correlation was assessed using a Spearman correlation with a Holm multiple testing correction. Total SNP distributions were compared using a Kruskal-Wallis test followed by pairwise Mann-Whitney U tests. Synonymous and nonsynonymous SNP distributions were compared with a Pearson's Chi-squared test followed by Bonferroni corrected Fisher's exact tests. The vancomycin and daptomycin MIC distributions between NRS70 VISA strains and NRS384 VISA strains were compared using two-sided two-sample Kolmogorov–Smirnov tests. To test if the linear relationship between vancomycin MIC and daptomycin MIC was dependent on background (NRS70 and NRS384), an Analysis of Covariance (ANCOVA) was performed.

To assess the effect of mutations in individual genes on vancomycin MIC, linear regression models were fitted using the *lm* function in R. *walK, prs, rpoB, rpoC, vraS, and yvqF* as binary variables (mutated or not) were used as predictors for log transformed vancomycin MICs. All mutations found in these genes were classified as nonsynonymous SNPs. The final model was chosen by backwards selection with the goal of minimizing Akaike information criterion (AIC) as in *Eyre et al. (2017)*.

To investigate differences in the effect sizes of SNPs on vancomycin MIC from background, standard linear models were fitted with interaction terms for SNP presence and background. In these models, strain background was included as a fixed-effects predictor because the mixed-effects version (with background as a vector random effect affecting the main and interaction effects of the SNPs) was not computationally stable. To study potential pairwise interactions between SNPs affecting the MIC of vancomycin, linear mixed-effects models were fit with presence or absence of SNPs as fixed-effects predictors (main and interaction effects) and with background (NRS70, NRS384) as a scalar random effect. Mixed-effects models with vector random effects (affecting intercept and slope of certain interactions) were also tested, but these models were not stable enough for convergence. Given the sparsity of the data matrix, the predictors were restricted to SNPs present in at least 10% of the strains. Predictors included were the six universally mutated genes (*walK, prs, rpoB, rpoC, vraS, yvqF*) and *sdrC*. All SNPs with the exception of SNPs in *sdrC*, which were all synonymous, were nonsynonymous. The response variable (vancomycin MIC) was log transformed. These models were also fitted for another response (log transformed growth rate as a proxy of fitness). Similar to the vancomycin models, pairwise SNP interactions were assessed with mixed-effects models and

SNP-background interactions with standard linear models. A Bonferroni corrected *p*-value (α = 0.00056) was used to assess test significance. R scripts used for these analyses are available at https://github.com/crsl4/staph-visa.

## Antimicrobial susceptibility testing

Antimicrobial susceptibility testing for daptomycin, vancomycin, and methicillin were performed according to Clinical Laboratory Standards Institute (CLSI) standards (*CLSI, 2013*). In brief, cultures were streaked from a frozen stock onto BHI agar and restreaked the next day. From the second day culture, colonies were resuspended in normal saline to a 0.5 McFarland standard. For plate-based testing, Etest for daptomycin and disc diffusion with cefoxitin, cultures were struck to create a lawn on cation-adjusted Mueller-Hinton broth (CAMHB) agar plates before application. Plates were incubated at 35 °C and read at 18 h. For broth microdilution (BMD) to determine strain vancomycin minimum inhibitory concentration (MIC), 0.5 McFarland standards were diluted 1:20 in normal saline, and 10 µL was added to 90 µL of CAMHB with the appropriate concentration of vancomycin. Plates were incubated at 35 °C without shaking and read at 24 h.

## RESULTS

### Patterns of acquisition of SNPs at VISA endpoints is influenced by genetic background

Three strains, representing major clinical lineages in the US: N315/NRS70 (*Kuroda et al., 2001*) (ST5/CC5), USA300-0114/NRS384 (*Diep et al., 2006*) (ST8/CC8) and MW2/NRS123 (*Baba et al., 2002*) (ST1/CC1) were experimentally evolved under selection pressure to achieve a vancomycin MIC of 4-10 µg/mL from an initial MIC of 1 µg/mL (Fig. 1). Selection of 120 of these mutants, 40 independent lines per genetic background, was achieved by sequentially raising the vancomycin concentration through serial passages of strains in BHI broth. Mutants were sequenced and compared to their isogenic parent using breseq (*Deatherage & Barrick, 2014*) to ascertain mutations acquired during vancomycin selection. Thirteen evolved strains are excluded from further analyses due to either low sequence quality or culture contamination. The final number of evolved VISA strains per background with high quality DNA sequences was 35 NRS70 strains, 34 NRS123 strains, and 38 NRS384 strains.

The majority of the genetic changes that occurred during the selection for vancomycin-intermediate resistance were single nucleotide polymorphisms (SNPs) and will be the focus of the rest of the analyses. Most deletions, insertions, and substitutions only occurred once (Tables S1–S6). Of note, prophage loss (StauST398-4 in NRS70, Pvl108 in NRS123, 23MRA in NRS384) was observed in four strains from each of the genetic backgrounds. Phage loss has been previously observed in vancomycin and daptomycin resistant isolates and has been associated with the induction of the SOS response by antibiotics (*Maiques et al., 2006*; *Boyle-Vavra et al., 2011*; *Machado et al., 2020*).

We found between 5 and 25 SNPs in evolved VISA strains compared to their respective parent strains, including non-synonymous mutations in several genes previously reported

**Table 1 Prevalence of mutations in six universally mutated genes by genetic background.** Mutations in these genes were called by breseq. The NRS70 walK prevalence calculation considers the presence of SNPs as well as inframe deletions. All other prevalence calculations are based solely on SNPs.

| Gene/Background | NRS70 | NRS123 | NRS384 |
|---|---|---|---|
| *walK* | 80% | 85.3% | 94.7% |
| *prs* | 5.7% | 5.9% | 2.6% |
| *rpoB* | 45.7% | 20.6% | 18.4% |
| *rpoC* | 34.3% | 20.6% | 10.5% |
| *vraS* | 17.1% | 23.6% | 39.5% |
| *yvqF* | 80% | 55.9% | 55.3% |

to be associated with VISA genes. As the genome sizes for NRS70, NRS123 and NRS384 are 2.81 MB, 2.82 MB, and 2.87 MB, respectively, we did not expect differences in genome size to significantly influence the number of mutations observed during the course of selection.

SNPs fell within 151 coding genes, and no gene was universally mutated in all strains, highlighting the existence of multiple pathways to intermediate vancomycin resistance in *S. aureus*. However, of the 151 genes, 121 genes had mutations in only one of the three backgrounds; 24 genes were mutated in two backgrounds; and only six genes (*walK, prs, rpoB, rpoC, vraS, yvqF*) had non-synonymous mutations in all three backgrounds. The six "universally mutated" genes have been previously implicated in VISA from sequencing of clinical isolates, but interestingly, some genes that had been previously reported did not acquire mutations frequently or in all three backgrounds. Notably, we did not find any *agrR* mutants, and *graRS* mutants were found in only NRS384 derived strains and *walR* (*Howden, Peleg & Stinear, 2013*) only in NRS70 and NRS123 derived strains.

The six universally mutated genes varied in their frequency across strains. On opposite sides of the spectrum, mutations in *prs* occurred in fewer than 10% of the evolved strains while mutations in *walK* occurred in 80–95% of the evolved VISA strains (Table 1). A Pearson's chi-squared test ($p < 2.2 \times 10^{-16}$) and post hoc Fisher's exact tests (all $p < 5.91 \times 10^{-03}$) determined that the proportion of strains differed significantly between each of these six genes. There was also evidence that some gene mutations occurred more frequently in some backgrounds than others. By fitting binomial generalized linear models (GLMs) to the individual gene mutation distributions with the genetic background as a predictor, proportions of strains with *rpoB* (NRS70 vs. NRS123, $p = 0.03$; NRS70 vs. NRS384, $p = 0.015$), *rpoC* (NRS70 vs. NRS384, $p = 0.019$), *vraS* (NRS70 vs. NRS384, $p = 0.04$), *and yvqF* (NRS70 vs. NRS123, $p = 0.035$; NRS70 vs. NRS384, $p = 0.028$) were found to be significantly influenced by genetic background. Between NRS70 and NRS123, five other genes were mutated in both backgrounds. Similarly, between NRS123 and NRS384, six other genes were mutated. Finally, between NRS70 and NRS384, 12 other genes were mutated (Table 2). The higher proportion of other shared genes between NRS70 and NRS384 may indicate that these two strains share more similar evolutionary

**Table 2 Genes with SNPs shared between two genetic backgrounds.**

| | NRS70 and NRS123 | NRS123 and NRS384 | NRS70 and NRS384 |
|---|---|---|---|
| Genes shared in at least 1 evolved isolate | WP_000101976.1 (*walR*) | WP_001081640.1 | WP_000631969.1 (*cysS*) |
| | WP_000035320.1 (*pdhA*) | (Cyclic-di-AMP phosphodiesterase) | WP_001060462.1 (*sdrC*) |
| | WP_001032833.1 (*gad*) | WP_000260117.1 (*pdhD*) | WP_001273060.1 (*tagG*) |
| | WP_000375864.1 | WP_001071136.1 (*mprF*) | WP_000491755.1 (*ykaA*) |
| | (HTH cro/C1-type domain-containing protein) | WP_000809131.1 (*cspA*) | WP_000120368.1 (*pitA*) |
| | WP_000251253.1 (UPF0374 protein) | WP_000048060.1 (*rpsU*) | WP_000431312.1 (*greA*) |
| | | WP_000830380.1 (*mgt*) | WP_000782121.1 (*prsA*) |
| | | | WP_000153535.1 (*vraR*) |
| | | | WP_000159960.1 (*pyrG*) |
| | | | WP_000347896.1 (*dacA*) |
| | | | WP_000004085.1 (*rpsQ*) |
| | | | WP_000008673.1 (*ureC*) |

paths toward vancomycin resistance, but mutations in these genes were rare, and differences were not statistically significant.

We examined the co-occurrence of mutations in the six universally mutated genes (Figs. 2A–2D). Mutations in *vraS* and *yvqF* were almost mutually exclusive (coefficient −0.8, $p < 0.0001$), with minor differences between genetic backgrounds (Figs. 2B–2D, NRS70 and NRS384: −0.9; NRS123: −0.6). NRS70 was the only background where we detected significant associations between other genes (Fig. 2B): *prs/vraS* (coefficient 0.5, $p = 0.011$) and *prs/yvqF* (coefficient 0.5, $p = 0.035$).

To examine the distribution of amino acid changes caused by SNPs, we focused on *yvqF* and *walK* due to the high frequency of mutation in our VISA strains (Table 1). For *yvqF*, there were 32 SNPs in 68 strains: 3 SNPs are found in all three backgrounds, 10 SNPs are found in two, and 19 SNPs in only one. Despite *walK* mutations (41 SNPs across 93 strains) being more common, there are no SNPs in common between the three backgrounds, and 31 SNPs are found in only one. Furthermore, the most common SNPs within each background for these two genes were found in different amino acid residues (*yvqF*: NRS70/A152V, NRS123/P126S, NRS384/P174A; *walK*: NRS70/E236D, NRS123/I29M, NRS384/Q216E).

In population genetics, there is a common a priori assumption that synonymous SNPs have little or no functional impact compared to non-synonymous. However, we noted some genes with high rates of synonymous mutations. Strikingly, *sdrC* was mutated in 7 out of 38 NRS384 VISA strains. The *sdr* locus encodes *sdrC, sdrD, sdrE* which are members of the repeat-rich microbial surface components recognizing adhesive matrix molecules (MSCRAMM) family (*Josefsson et al., 1998*). These genes are not always conserved together (*Liu et al., 2015*), and their presence has been associated with bone infection (*Sabat et al., 2006*), resistance to host immunity (*Sitkiewicz, Babiak & Hryniewicz, 2011*; *Askarian et al., 2017*), biofilm formation (*Barbu et al., 2014*), and host-cell adhesion (*Corrigan, Miajlovic & Foster, 2009*; *Cheng et al., 2009*; *Barbu et al., 2010*; *Askarian et al., 2016*). *Belikova et al. (2020)* showed that in USA300 *sdr* genes undergo frequent within-genome recombination during growth in lab conditions and during infection.

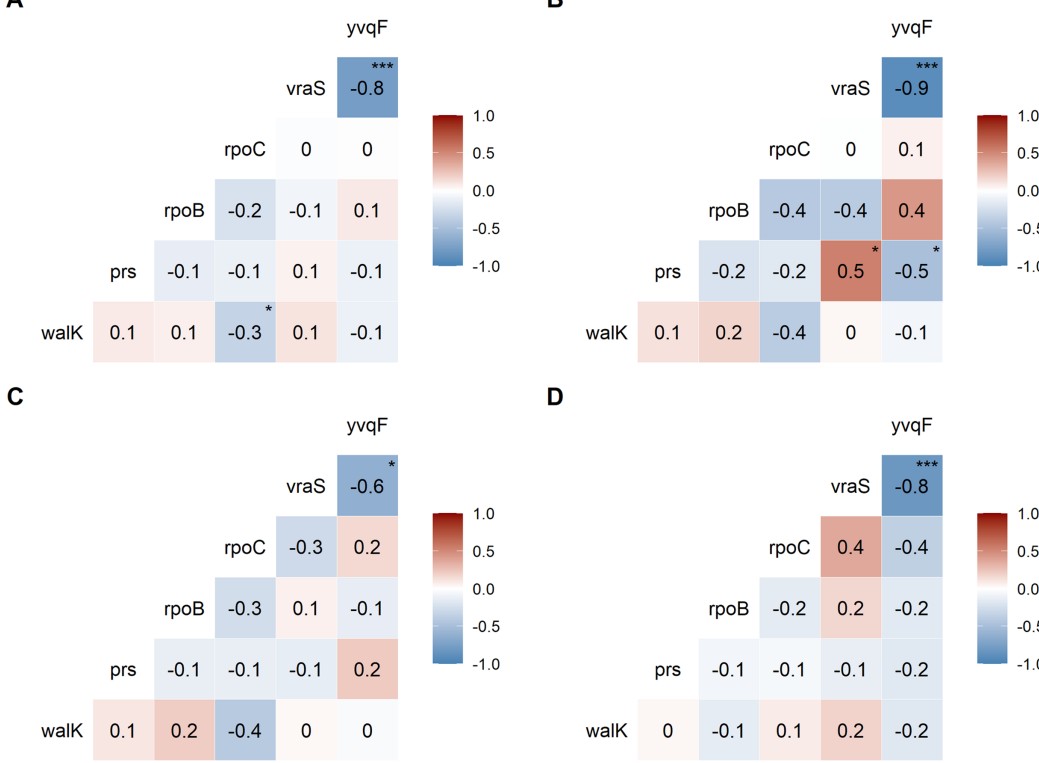

**Figure 2 Correlation of mutations between VISA genes.** The numbers are correlation coefficients between genes based on the presence of non-synonymous SNPs or frameshift deletions in the same strain (A) Correlation matrix of all evolved VISA strains. (B) Correlation matrix of all NRS70 evolved VISA strains. (C) Correlation matrix of all NRS123 evolved VISA strains. (D) Correlation matrix of all NRS384 evolved VISA strains. Significance codes: '***' <0.0001 '**' <0.001 '*' <0.05.

The high number of synonymous mutations may be the result of repeats expanding or collapsing in evolved strains, causing base-calling artefacts when mapped onto the reference genomes. Ultimately, long-read sequencing approaches are needed to deconvolute these complex structures, but there is a strong possibility that base-calling artefacts inflated synonymous mutation numbers in some strains. We also noted that there were significant differences between backgrounds in the total number of mutations that accrued during the evolution experiments ($p = 1.1 \times 10^{-5}$). NRS123 VISA (median 3 mutations) strains had fewer mutations than NRS70 (median 4, $p = 0.004$) and NRS384 (median 5, $p = 2.8 \times 10^{-5}$) VISA strains (Fig. 3). Furthermore, the number of synonymous SNPs relative to nonsynonymous SNPs was not the same (Table 3, Pearson chi square $p = 1.8 \times 10^{-5}$, NRS70 vs. NRS123 $p = 0.046$, NRS70 vs. NRS384 $p = 0.038$, NRS123 vs. NRS384 $p = 1.0 \times 10^{-7}$). However, when excluding *sdr* mutations, NRS70 and NRS123 are no longer statistically significantly different, but NRS384 remained statistically different from NRS70 and NRS123 (Table 3, Pearson chi square $p = 0.00021$, NRS70 vs. NRS384 $p = 0.014$, NRS123 vs. NRS384 $p = 0.0011$). Furthermore, the number of synonymous SNPs relative to nonsynonymous SNPs was not the same (Table 3, Pearson chi square $p = 1.8 \times$

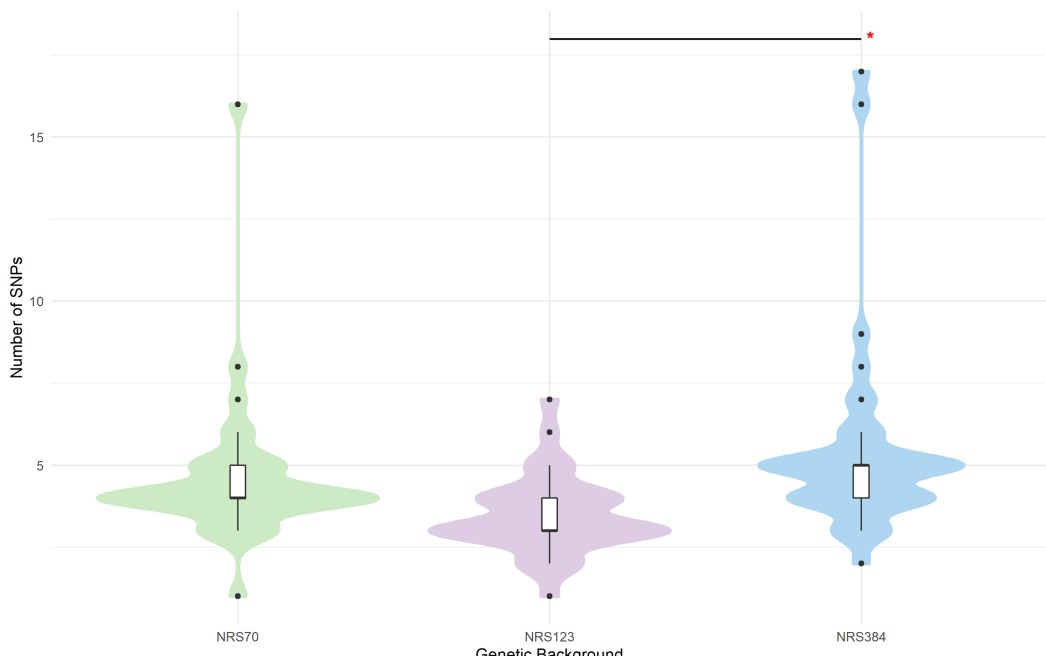

**Figure 3 SNP distributions of VISA evolved strains differ by genetic background.** Mutations were called with breseq. Duplicate *sdr* mutations have been removed. Mode and median of SNPs per background: NRS70 (4), NRS123 (3), NRS384 (5). Significance codes: An asterisk (*) designates $p < 0.06$.

**Table 3 Synonymous SNPs and nonsynonymous SNPs by genetic background.**

|         | Synonymous | Nonsynonymous |
|---------|-----------|---------------|
| **NRS70**  | 19 (10)   | 149           |
| **NRS123** | 4         | 114           |
| **NRS384** | 45 (32)   | 166           |

Note:
Numbers given are the sum of the SNPs of all strains in that background (NRS70: 35 strains, NRS123: 34 strains, NRS384: 38 strains). The number of mutations when excluding sdr operon mutations is given in parenthesis.

$10^{-5}$, NRS70 vs. NRS123 $p = 0.046$, NRS70 vs. NRS384 $p = 0.038$, NRS123 vs. NRS384 $p = 1.0 \times 10^{-7}$).

## Resistance phenotypes of evolved VISA strains

All evolved VISA strains at the endpoint of the iterative selection were able to grow in 8 µg/mL vancomycin BHI broth. However, to accurately measure the level of resistance, we performed BMD on NRS70 and NRS384 strains according to CLSI standards (unfortunately, all NRS123 frozen cultures were accidentally discarded while in −80 °C storage before we could test them). All endpoint NRS70 and NRS384 strains had MICs within the range to be considered VISA (Fig. 4A), but some (20 NRS70 and 28 NRS384 strains) were found to be below 8 µg/mL. The NRS70 VISA strain MICs ranged from 4 to 10 µg/mL, with a median and mode of 6 µg/mL, while the NRS384 VISA strain MICs ranged from 4 to 8 µg/mL, with a median and mode of 6 µg/mL. The difference in the

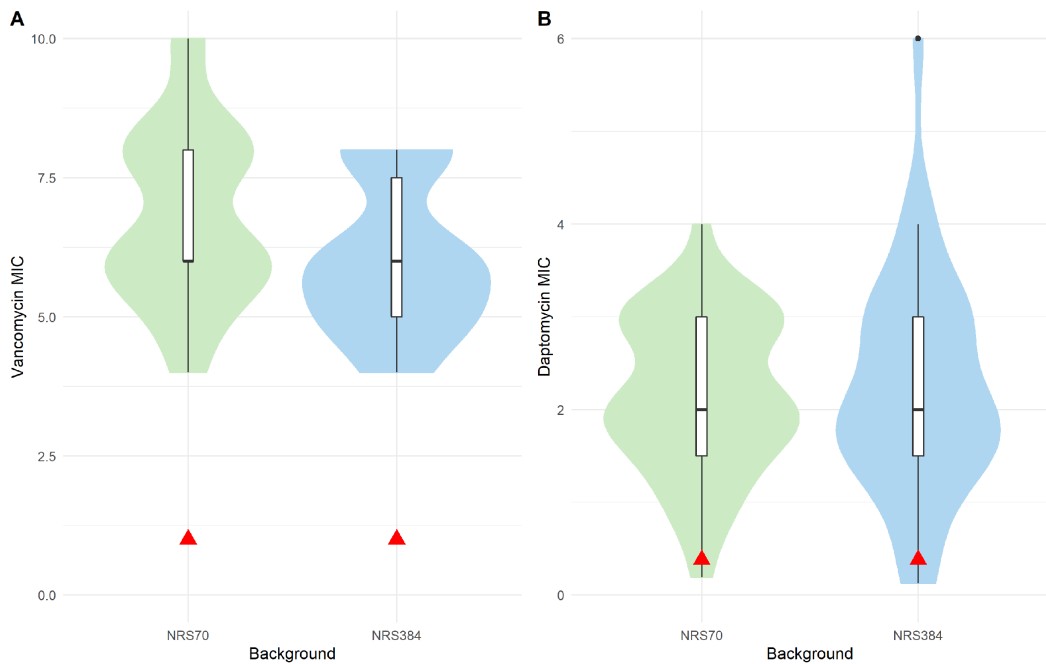

**Figure 4 Antibiotic MIC distributions of NRS70 and NRS384 evolved strains.** Vancomycin MICs were determined by BMD and daptomycin MICs by Etest. (A) Vancomycin MIC distributions. The comparison of the two distributions by a Kolmogorov–Smirnov test was not statistically significant ($p = 0.3236$). Parental MICs are indicated by a red triangle at 1 μg/ml. (B) Daptomycin MIC distributions. The comparison of the two distributions by a Kolmogorov–Smirnov test was not statistically significant ($p = 0.9996$). Parental MICs are indicated by a red triangle at 0.38 ug/ml.

MIC distributions was not statistically significant between NRS70 and NRS384 strains ($p = 0.323$). Mutated genes and/or mutated gene patterns associated with isolates with the highest level of vancomycin resistance were not appreciably different than those found in isolates with a lower resistance, likely indicating that individual mutations played a large role in the resistance achieved versus genes overall.

The cross-resistance of vancomycin and daptomycin in VISA strains has been well-studied (*Cui et al., 2006*; *Nam et al., 2018*). NRS70 and NRS384 endpoint strains were thus tested for development of daptomycin resistance by Etest. Parental strains had daptomycin MICs of 0.38 μg/mL. Thirty of the 35 NRS70 strains became resistant, and 33 of the 38 NRS384 strains became resistant (Fig. 4B). As observed with the vancomycin resistance genes, there was no clear mutational pattern associated with higher daptomycin resistance. The daptomycin distributions between NRS70 and NRS384 were not statistically different ($p = 1$), however, only NRS384 strains had a daptomycin MIC of 6 μg/mL versus 4 μg/mL for NRS70 strains. Greater levels of vancomycin resistance correlated to greater levels of daptomycin resistance (Fig. 5, $p = 0.006$). However, the regressions for each background were not significantly different ($p = 0.822$).

Collateral sensitivity with beta-lactams is a less frequently reported phenomenon in VISA strains (*Sieradzki & Tomasz, 1999*; *Mwangi et al., 2007*; *Barber et al., 2014*). Within our evolved NRS70 and NRS384 strains, only two NRS70 strains became methicillin
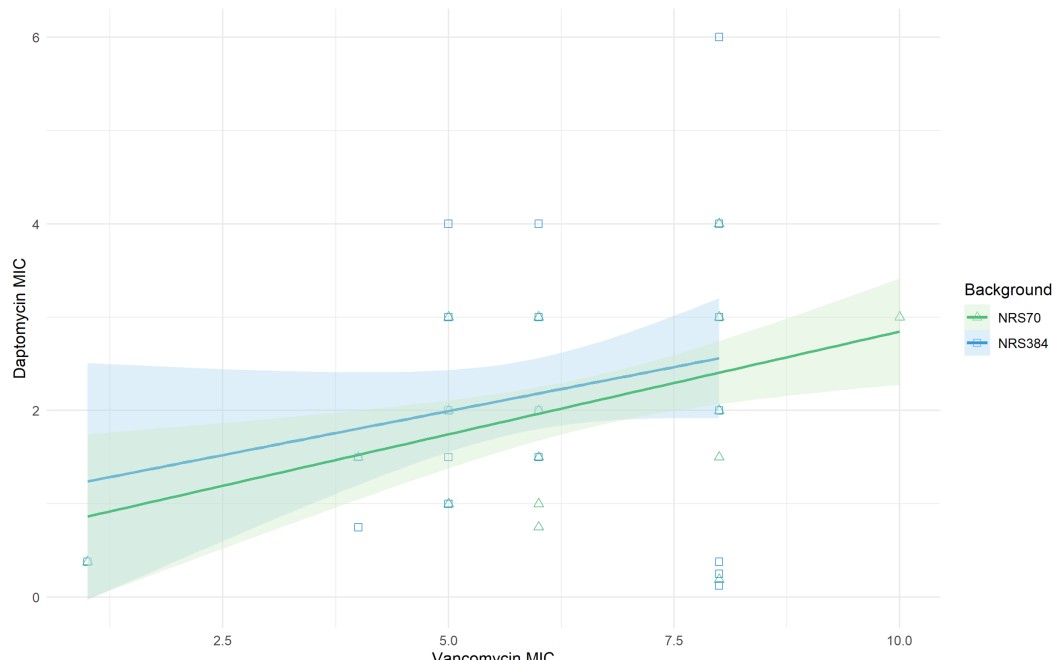

**Figure 5 Linear relationship between vancomycin MIC and daptomycin MIC.** Vancomycin MICs were determined by BMD and daptomycin MICs by Etest. Linear regression of daptomycin MIC with vancomycin MIC was statistically significant ($p$ = 0.006). The comparison of the background specific linear relationships between vancomycin and daptomycin by an Analysis of Covariance (ANCOVA) was not statistically significant ($p$ = 0.822). Overall: Adjusted R-squared: 0.08903 ($p$ = 0.005); N70: Adjusted R-squared: 0.1866 ($p$ = 0.005); N384: Adjusted R-squared: 0.03534 ($p$ = 0.13).

sensitive. Known mutations for beta-lactam sensitivity in a MRSA background such as loss of SCC*mec* or inactivation of *mecA* were not found, and there the mechanism for collateral sensitivity in these strains was unknown.

## Fitness decreases during evolution of higher resistance but is modulated by genetic background

We examined the fitness of intermediate and endpoint strains of each NRS70 and NRS384 experimentally evolved lineage using maximum growth rate in batch culture log(r) as an indicator. Isolates from later stages of selection, able to tolerate greater concentration of vancomycin, generally equated to lower fitness (Figs. 6A–6B). Looking at the fitness distributions at each selection stage (4 μg/mL, 6 μg/mL, 8 μg/mL) between NRS70 and NRS384, there was a striking difference at 4 μg/mL ($p$ = $7.08 \times 10^{-6}$) and 6 μg/mL ($p$ = $1.86 \times 10^{-6}$), which disappeared at 8 μg/mL ($p$ = 0.273) (Figs. 7A–7C). Thus, NRS70 strains appeared to be able to tolerate lower levels of vancomycin resistance with lower fitness cost compared to NRS384 strains. Linear mixed models with SNP presence/absence as fixed effects and background as a scalar random effect were used to determine interaction effects between individual SNPs and log(r). However, no results were statistically significant after Bonferroni correction. The strongest interaction was between *sdrC* and *rpoC* whereby mutations in *sdrC* had a negative effect on growth rate in the
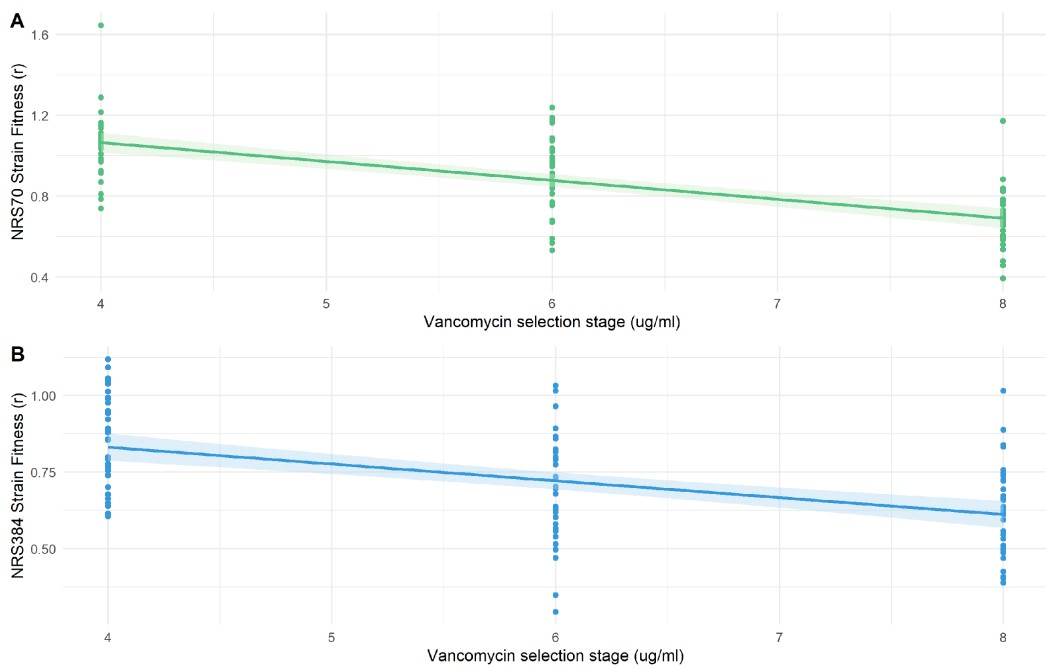

**Figure 6 Fitness of NRS70 and NRS384 evolved strains.** Strains were grown in duplicate in a 96-well plate beginning at an OD < 0.1 and grown for 24 h at 37 °C with constant shaking in a Biotek Eon Microplate Spectrophotometer, with OD measurements every 10 min. To assess growth-curves, OD readings were imported into R and maximal growth rate (r) calculated using the growthcurver package. Fitness was calculated as the ratio between the average r of each evolved strain to the average r of the parent strain. (a) Fitness (r) of NRS70 evolved strains plotted against the concentration of vancomycin that the strains were evolved. NRS70-7 at vancomycin selection stage 6 and 8 and NRS70-14 at vancomycin selection stage 6 were excluded due to lack of growth or too large a variance between replicates. (b) Fitness (r) progression of NRS384 evolved strains.

presence of *rpoC* mutation from a previous positive effect (uncorrected $p = 0.023$, Fig. S1), but as noted above, the caveat is that *sdrC* mutations may be surrogates for more complex recombination events.

## Genetic predictors of vancomycin MIC

To investigate the interactions of mutation and *S. aureus* strain background to vancomycin resistance, we used two modelling approaches. In the first, we used multiple linear regression models to determine which mutated genes were most predictive of vancomycin resistance. To reduce the number of parameters and for ease of comparison, only the six universally mutated genes were used in the initial models (Table 4). To make the models more meaningful, parameters were evaluated for inclusion in the final models using AIC (*Eyre et al., 2017*). In the final model for NRS70, *walK* (coefficient estimate 0.298), *rpoC* (0.228), *vraS* (0.989), and *yvqF* (1.058) were statistically significantly associated with vancomycin resistance, defined as log(MIC) ($R^2 = 0.583$, $p = 2.15 \times 10^{-6}$). In comparison, only *walK* (coefficient estimate 0.633) by itself was statistically significant in the final model of NRS384 ($R^2 = 0.313$, $p = 9.95 \times 10^{-4}$), although *vraS* (0.340) and *yvqF* (0.378) were included for improved model fit.
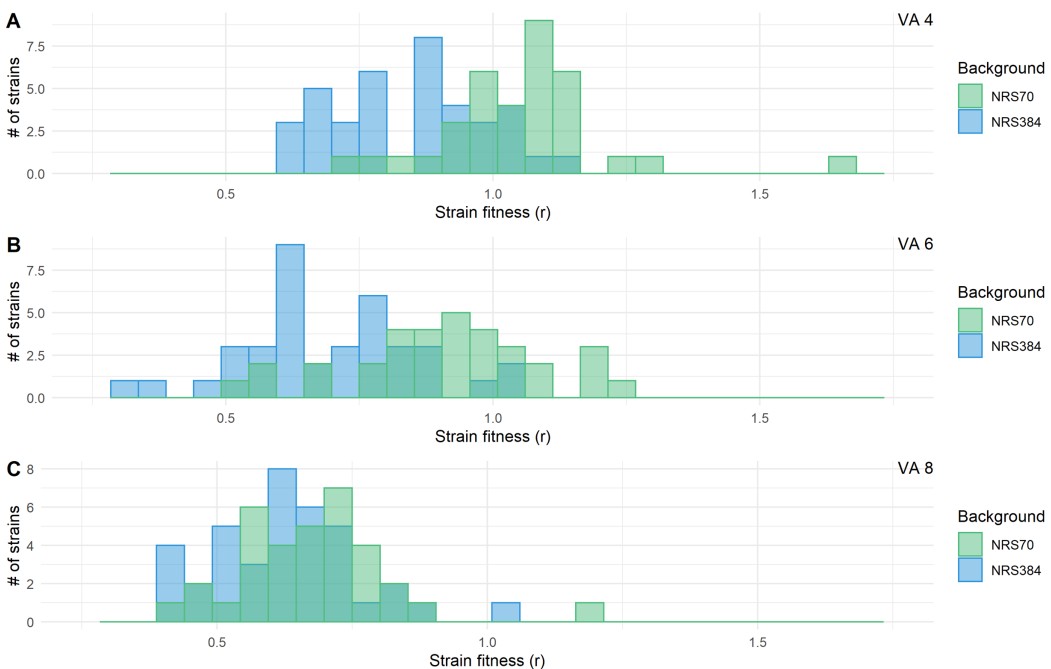

**Figure 7 Fitness distributions at each stage of vancomycin selection for NRS70 and NRS384 strains.** MICs were determined by BMD. Strains were grown in duplicate in a 96-well plate beginning at an OD < 0.1 and grown for 24 h at 37 °C with constant shaking in a Biotek Eon Microplate Spectrophotometer, with OD measurements every 10 min. To assess growth-curves, OD readings were imported into R and maximal growth rate (r) calculated using the growthcurver package. Fitness was calculated as the ratio between the average r of each evolved strain to the average r of the parent strain. Fitness distributions between NRS70 evolved VISA strains and NRS384 evolved VISA strains were compared using a two-sided two-sample Kolmogorov–Smirnov test. (A) Vancomycin selection stage 4 µg/mL. (B) Vancomycin selection stage 6 µg/mL. (C) Vancomycin selection stage 8 µg/mL. NRS70-7 at vancomycin selection stage 6 and 8 and NRS70-14 at vancomycin selection stage 6 were excluded due to lack of growth or too large a variance between replicates.

**Table 4 Linear regression models of vancomycin MIC.** An initial model was fitted using walK, prs, rpoB, rpoC, vraS, and yvqF as binary (mutated or not) predictors for log transformed vancomycin MICs. The predictors in the final were chosen by backwards selection.

| Background | Gene | Estimate (initial) | p (initial) | Estimate (final) | p (final) |
|---|---|---|---|---|---|
| **NRS70** | *walK* | 0.288 | 0.015 | 0.298 | 0.012 |
| | *prs* | 0.198 | 0.389 | – | – |
| | *rpoB* | 0.132 | 0.254 | – | – |
| | *rpoC* | 0.309 | 0.012 | 0.228 | 0.026 |
| | *vraS* | 0.901 | 0.0005 | 0.989 | $<10^{-4}$ |
| | *yvqF* | 0.956 | $<10^{-4}$ | 1.058 | $<10^{-5}$ |
| **NRS384** | *walK* | 0.629 | 0.004 | 0.633 | 0.002 |
| | *prs* | 0.255 | 0.515 | – | – |
| | *rpoB* | 0.103 | 0.460 | – | – |
| | *rpoC* | −0.050 | 0.790 | – | – |
| | *vraS* | 0.412 | 0.129 | 0.340 | 0.096 |
| | *yvqF* | 0.450 | 0.073 | 0.378 | 0.053 |

In the second approach, we fit linear regression models to determine if the effect sizes of SNPs on log(MIC) differed by background. No results were statistically significant after Bonferroni correction, but there was a suggestive result that *rpoC* may contribute positively to vancomycin MIC in NRS70 strains but not NRS384 strains (uncorrected $p = 0.028$, Fig. S2), which is consistent with our previous linear regression models. Linear mixed models were used to determine interaction effects between SNPs. No results were statistically significant after Bonferroni correction, but there was a suggestive result that the effect of a *sdrC* mutation (either increasing or decreasing vancomycin MIC) depended on *walK* (uncorrected $p = 0.0472$, Fig. S3). Overall, the models indicated that while the six universally mutated core genes were broadly responsible for vancomycin resistance in NRS70 and NRS384, they were of differential importance within each background. In addition, we did not find strong evidence for pairwise interaction between SNPs influencing MIC.

## DISCUSSION

In this project we looked for evidence of strain specificity in patterns of genetic change responsible for transition to vancomycin intermediate resistance in *S. aureus*. We found evidence that mutations in the same gene had differential effects on strain fitness and MIC depending on the genetic background. There were also statistically significant correlations in the patterns of co-occurrence of mutations linked to VISA. We did not probe the mechanisms of the putative epistatic interactions in this study. It is possible that, since many VISA-linked genes affect global cellular regulatory pathways, there is cross-talk with fixed mutations in the strain background that set global state changes (*Priest et al., 2012*). The strains also differ in their accessory gene content (*Lindsay et al., 2006*; *McCarthy & Lindsay, 2010*; *Aanensen et al., 2016*), which may also affect gene expression and protein interactions responsible for the VISA phenotype. This work adds to a growing body of literature on the importance of strain background in this pathogen. S. *aureus* lineages have been shown to be heterogeneous in several significant clinically relevant phenotypes such as toxin production, biofilm formation, and host immune resistance (*King et al., 2016*; *Su et al., 2020*). Epistatic interactions between antibiotic resistance and toxicity have also been explored (*Yokoyama et al., 2018*).

VISA was found to be linked to mutations (primarily SNPs) in a limited number of genes (e.g., *walKR*, *rpoB/C*, *vraTSR*). Six previously implicated VISA-associated genes (*walK*, *prs*, *rpoB*, *rpoC*, *vraS*, *yvqF*) (*Howden et al., 2010*) were mutated in our three genetic backgrounds. *walK* followed by *yvqF* were the most commonly seen mutations. For *rpoB*, *rpoC*, *vraS*, and *yvqF*, NRS70 and NRS384 mutation incidence were most different, and NRS123 mutation incidence was intermediate between the others. These results may be partially explained by the closer genetic relationship between CC1 and CC8 than CC5 and CC8 (*Petit & Read, 2018*). Other genes were found to have been mutated in two of the three genetic backgrounds and may have a role in vancomycin resistance or act as compensatory mutations. Several are already validated as being associated with vancomycin resistance (*walR* (*Howden et al., 2011*), *vraR* (*Kato et al., 2010*; *Baek et al., 2017*; *Asadpour & Ghazanfari, 2019*), *mprF* (*Ruzin et al., 2003*; *Chen et al., 2018*)). NRS70

and NRS384 backgrounds had more of these genes in common than with NRS123, indicating that NRS70 and NRS384 strains may experience more similar selective pressures in the larger mutational landscape outside of the previously discussed six genes.

The overall vancomycin resistance levels achieved after experimental evolution in NRS70 and NRS384 VISA strains were similar, suggesting both backgrounds are equally capable of achieving lower and higher levels of vancomycin intermediate resistance. Linear regression demonstrated that genetic predictors of vancomycin resistance differed between backgrounds. *walK, rpoC, vraS,* and *yvqF* were all significantly associated with vancomycin MICs in NRS70, with *vraS* and *yvqF* contributing most significantly, but only *walK* was significant in NRS384. As NRS384 strains carried more mutations than the other two backgrounds, this may indicate that while a subset of the six universally mutated genes (*walK, prs, rpoB, rpoC, vraS, yvqF*) are necessary for vancomycin resistance, other genes may play a significant role in determining the level of vancomycin resistance achieved in NRS384 VISA strains but not NRS70 strains.

We found that vancomycin-intermediate resistance imposed a fitness cost (measured by maximal growth rate compared to parent strain) to *S. aureus* that linearly scaled with the level of vancomycin resistance in most cases. Fitness distributions between these two genetic backgrounds were markedly different at stages 4 µg/mL and 6 µg/mL, with vancomycin resistance imposing less of a fitness cost on NRS70 VISA strains than NRS384 VISA strains. This may have clinical implications as it suggests that NRS70 *S. aureus* strains may evolve vancomycin resistance quickly, not be rapidly selected against due to growth defects, and may reach fixation if vancomycin concentrations are sustained and lead to the extinction of the ancestral vancomycin susceptible strain.

The laboratory evolution approach used in this study gave us the power to evolve VISA multiple times independently and achieve statistical significance using general linear models. With these data, we were able to investigate the effect of different SNPs with the strain background as a fixed effect. However, we cannot draw strong conclusions about clinical relevance as clinical VISA strains are the result of within-host evolution, and immune pressure is not present within our experimental design. Other factors that are present in human infection that were omitted in our experiment include but are not limited to nutrient limiting conditions and microenvironments. These undoubtedly reduce the mutational space *S. aureus* has available for adaption. In addition, the mutations in our studies were grouped by gene for simplicity, but likely not all mutations in VISA genes contribute equally to vancomycin resistance, and some may have no effect. The sheer number of mutations makes it impossible to recapitulate these mutations individually. We did not run control evolution experiments without vancomycin, so we cannot say whether some of the mutations in the most common six gene were due to random genetic drift, although it is well established that VISA strains do not evolve from VSSA in the laboratory in the absence of antibiotic selection. Another study design limitation was that fitness determinations by measuring maximal growth rate do not always correlate with competitive indices of evolved strains to ancestor strains and have been shown to be sensitive to environmental conditions. Thus, our measured fitness values may not accurately reflect within-host fitness, especially in environments with competing microbes.

The results of this study offer some encouragement that development of VISA strains may be predicted with reasonable accuracy directly from the genome sequence, an important future technique in diagnostic clinical microbiology (*Su, Satola & Read, 2019*). This is because the most common VISA mutations occurred in all three backgrounds, suggesting a "dictionary" of common mutations could be compiled. However, the finding that individual gene contribution to MIC may be background dependent (Table 4) suggests that prediction of the level of resistance in strains from different clonal complexes may be a much harder problem. In order to obtain the data to make exhaustive catalogs of VISA mutations and their epistatic dependencies or to build an accurate classifier, much more extensive versions of the experiment approach outlined here need to be performed, with a greater number of representative genetic backgrounds.

## CONCLUSIONS

We found that there was a complex relationship between genetic background and VISA. There was clear evidence for parallel evolution of VISA through mutations in a common set of genes across strains (especially: *walK, prs, rpoB, rpoC, vraS, yvqF*). Some of these genes (particularly *yvqF* and *vraS*) showed negative epistasis in all strains. However, we found evidence the effects of mutations on vancomycin MIC and the relationship between fitness and MIC were dependent on strain background. These results indicate that prediction of the levels of vancomycin resistance based on genome sequence may require extensive databases of mutants from different genetic backgrounds.

## ACKNOWLEDGEMENTS

We would like to thank Jon Moller and Robert Petit for discussion and comments on the manuscript as well as Sujith Swarna for help with the experimental evolution experiments.

### Funding

Timothy D. Read was supported by the National Institute of Allergy and Infectious Diseases (NIAID) award AI121860. Michelle Su was supported by the Antimicrobial Resistance and Therapeutic Discovery Training Program funded by NIAID T32 award AI106699-05. There was no additional external funding received for this study. The funders had no role in study design, data collection and analysis, decision to publish, or preparation of the manuscript.

### Grant Disclosures

The following grant information was disclosed by the authors:
National Institute of Allergy and Infectious Diseases (NIAID): AI121860.
Antimicrobial Resistance and Therapeutic Discovery Training Program: NIAID T32 award AI106699-05.

### Competing Interests

Timothy D. Read is an Academic Editor for PeerJ.

## Author Contributions

- Michelle Su conceived and designed the experiments, performed the experiments, analyzed the data, prepared figures and/or tables, authored or reviewed drafts of the paper, and approved the final draft.
- Michelle H. Davis performed the experiments, analyzed the data, authored or reviewed drafts of the paper, and approved the final draft.
- Jessica Peterson performed the experiments, analyzed the data, authored or reviewed drafts of the paper, and approved the final draft.
- Claudia Solis-Lemus analyzed the data, prepared figures and/or tables, authored or reviewed drafts of the paper, and approved the final draft.
- Sarah W. Satola analyzed the data, authored or reviewed drafts of the paper, and approved the final draft.
- Timothy D. Read conceived and designed the experiments, analyzed the data, authored or reviewed drafts of the paper, and approved the final draft.

## DNA Deposition

The following information was supplied regarding the deposition of DNA sequences:
Raw read data are available in the NCBI Short Read Archive: PRJNA525705.

## Data Availability

The raw data are available in the Supplemental Files. Antimicrobial susceptibility testing results for VISA strains are given for vancomycin, daptomycin, and cefoxitin. These data were used for analyses regarding MIC distribution and fitness comparisons. The raw data used for fitness/growth rate (r) analyses is given: OD measurements every 10 min for 24 h.

## Supplemental Information

Supplemental information for this article can be found online at http://dx.doi.org/10.7717/peerj.11764#supplemental-information.

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
