# Peer review of "Effect of genetic background on the evolution of Vancomycin-Intermediate Staphylococcus aureus (VISA)"

_PeerJ, doi:10.7717/peerj.11764_

## Round 0.1 · original submission · Major Revisions

Both reviewers were very supportive of this research, but both also identified a need for more experimental details . I concur and ask that you respond carefully to each of the reviewers' suggestions and critiques. Reviewer 1 also asked for inclusion of a media-only control arm, or at least an acknowledgment of experimental design limitations in the discussion.

·

Basic reporting

No comment.

Experimental design

The research question is well defined, relevant and meaningful. My only comment on experimental design is that these kinds of experimental evolution experiments should ideally include a "media only" arm, where bacteria are passaged without antibiotic to allow for the identification of mutations that arose due to passaging in the growth media, rather than antibiotic selection. Alternately, the parent strain can be "pre-adapted" to the growth media before the selection experiment starts. If the authors haven't done either of these, then there needs to be more explicit mention in the Discussion of how mutations observed could be due to adaptation to experimental conditions, rather than antibiotic selection.

Validity of the findings

No comment.

Additional comments

The manuscript by Su et al. describes a thorough investigation into patterns of genetic adaptation that arise during the evolution of VISA. Besides my comment above about the experimental design, I have a few additional comments:

1. There should be more attention paid to the actual mutations detected in each of the genes of focus in this study. The mutations themselves should be summarized somewhere, such as a supplemental table. In reading the paper I got the sense that all mutations in a given gene were being treated equally by the authors, but is this really the case? Presumably mutations in different regions/domains of a given gene can have different downstream effects. While a variety of different mutations were observed in each gene, did they tend to cluster in particular regions/domains? This information is useful for thinking about how mutations in these genes exert an effect on VISA phenotype, and should therefore be included. Finally, I'm a bit confused about how conclusions regarding genetic background effects can be made when the actual mutations in the genes are different. Unless the authors are confident that two different non-synonymous SNPs in a particular gene have the same functional effect, I would suggest softening the conclusions regarding epistasis except in cases where allele-level parallelism (rather than just gene-level parallelism) is observed.

2. Are the genome sizes of the parent strains very different from one another? If so this could effect the number of mutations observed during the course of selection. Either way parental strain genome sizes should be reported.

3. Tables 1-5 should be moved to supplemental material, since indels were not a focus of the study.

4. It would be helpful to see the MICs of the parent strains indicated somehow on Figure 3.

5. What are the actual mutations in the isolates with the highest Daptomycin and Vancomycin MICs in Figure 4? Why do the authors think that these mutations have the greatest phenotypic effect?

6. Lines 146-148 in the methods should also be stated in the Figure legends for Figures 5 and 6.

7. Lines 25-27 in the abstract were confusing to me. I would suggest revising to "We used an iterative method to experimentally evolve three important MRSA strains to generate a library of 120 VISA isolates."

Reviewer 2 ·

Basic reporting

The manuscript it well written and provides a clear description of both methods and results. The introduction is well structured and provides a suitable context for the study. The review of the literature is appropriate for the topic of vancomycin resistance.

The structure is consistent with the scope of PeerJ. The figures are of good quality and clearly labelled. A figure 1 with a summary of the experimental evolution approach would be very helpful (see below).

There are two excel tables with raw data are on phenotypic testing of evolved isolates, however the authors should also provide a table with all mutations detected (eg breseq output in tabular form).

Experimental design

This is an original genomic study of experimentally evolved vancomycin-resistant S. aureus strains. It fits the scope of the journal. The research question (“What is the impact of the clonal background of MRSA on the acquisition of vancomycin resistance in terms of mutations, fitness cost, ability to become of resistance and co-resistance to other antibiotics?”) is clearly stated and fill a knowledge gap in the field.

The investigation is rigorous, and the methods applied in terms of resistance selection, genomic analysis and phenotypic testing are state-of-the art. More detail should be provided on the genomic analysis (eg QC of reads, were mutations filtered / manually validated?).

Regarding the statistical analysis, the authors combined different approaches to assess the impact of background, SNPs on the phenotypes: first, they used non-parametric test to compare backgrounds in terms of vancomycin MIC, growth rate, number of mutations and prevalence of mutations in the top mutated genes (the latter with Bonferroni correction); second, they fitted a binomial generalised linear model (logistic regression) to assess the impact of background on the prevalence of mutations in the top mutated genes. Finally, the fitted linear models where vancomycin MIC was the response variable and the top mutated genes were predictors (as binary variable mutated - non mutated).

Overall, the approach used seems appropriate, however, the statistical analysis is quite complex and I suggest providing a more structured description in the methods session, for example separating simple comparisons between groups and more complex models. It would also be good if the authors could share the code used, at least for the complex linear regression models.

Validity of the findings

The main findings of the study are :
1. Vancomycin resistance selection in CC8 background appears to be more difficult with a higher number of selection steps, higher number of mutation and higher fitness cost
2. While there are mutations in 151 genes among 107 evolved strains, six genes (walK, prs, rpoB, rpoC, vraS, yvqF/vraT) have a strong signature of convergence
These findings are supported by the data and are also consistent with what is known about the genetic determinants of VISA so far. However, this is not a mere replication study, since it explores the relationship between mutations and clonal background and also, by analysis 107 independently evolved strains, uses an approach that increases the chance of identifying convergent mutations and that allows to rank the importance of these mutations / mutated genes.’

The conclusions are well stated and there is a clear recognition that, while these findings might inform the use of genomics for the prediction of vancomycin resistance, data from experimental data alone are not sufficient for prediction of resistance.

Additional comments

1. While the six “universally mutated genes” captures more attention, I think that the authors should explore the remaining 145 mutated genes a little bit more. I’m impressed by this high number: how many of these have been previously linked to vancomycin / antibiotic resistance? Are there compensatory mutations?
2. Did the authors perform an epistasis analysis of the 151 mutated genes? This should give a good opportunity to explore the issue of epistasis in antibiotic resistance as stated in the introduction
3. Again to investigate epistasis and compensatory mutations a bit further, it might be useful to sequence strains obtained at an intermediate stage of the resistance selection
4. Please add a figure 1 with an overview of the experimental approach: resistance selection steps, number of independent experiments, choice of strains for downstream phenotypic and genomic analyses
5. Line 147: you mention the “average r of each strain”: were growth curves measures performed in replicates?
6. Line 151: see comments above for the description of the statistical analysis.
7. Line 219: truncations are often found in experimental evolution: what was the proportion here? Did you look at intergenic mutations to investigate whether they occurred in promoter regions of known VISA determinants?
8. Line 227 and figure 2: there are a couple of outliers among the NRS70 and NRS384 mutants that have a very high number of SNPs (> 20). What is the reason for this? Were the alignments checked manually to exclude false positive calls (eg in repetitive regions)?
9. Line 229: was there convergence among mutations / positions or mutated regions within the top mutated genes?
10. Line 320: Which genes were most frequently mutated in evolved strains with co-resistance to daptomycin?
11. Line 331: given the accumulation of mutations in some strains, it would be interesting to test whether re-sensitisation to beta-lactams was related to compensatory mutations
12. Line 354: I appreciate the use of modelling to investigate the impact of single mutated genes and genetic background on the final vancomycin MIC. Here add a comment on stats approach
13. Please add a table with the list of mutations / large deletions / insertions identified using breseq

---

## Round 0.2 · Minor Revisions

There are two items for your attention. Reviewer 1 asked that you review the manuscript to remove all typos and minor grammatical errors. Reviewer 2 asks that you provide more information so readers can better understand the content of Tables S2-S6.

·

Basic reporting

No comment.

Experimental design

No comment.

Validity of the findings

No comment.

Additional comments

The authors have addressed all of my comments in their revised manuscript.

Reviewer 2 ·

Basic reporting

The authors have appropriately responded to the reviewer’s comments and the new version of the manuscript is now clearer and provides more information on statistical methods and on the details of the variants identified.

Experimental design

No new experiments have been performed for this revision; however, the authors have now performed a genome-wide epistasis analysis using SpydrPick, a tool for bacterial genome-wide epistasis studies (GWES). It is plausible that the dataset was too small for a GWES.

I agree with the authors that an epistasis analysis would require further data and investigations that are beyond the scope of this manuscript.

Validity of the findings

No comments, since no new findings have been added

Additional comments

1. Please provide a description of the supplementary data. I’m not sure on what supplemental tables 2-6 represent

---

## Round 0.3 · accepted · Accept

I am satisfied with the authors' responses to the remaining reviewer issues.